# Nonthermal Plasma Synthesis of Metallic Ti Nanocrystals

**DOI:** 10.3390/nano14030264

**Published:** 2024-01-26

**Authors:** Qiaomiao Tu, David L. Poerschke, Uwe R. Kortshagen

**Affiliations:** 1Department of Chemical Engineering and Materials Science, University of Minnesota, Minneapolis, MN 55455, USA; tu000007@umn.edu; 2Department of Mechanical Engineering, University of Minnesota, Minneapolis, MN 55455, USA

**Keywords:** titanium, nanoparticles, nonthermal plasma, FCC

## Abstract

Nanoscale metallic titanium (Ti) offers unique energetic and biocompatible characteristics for the aerospace and biomedical industries. A rapid and sustainable method to form purified Ti nanocrystals is still in demand due to their high oxygen affinity. Herein, we report the production of highly purified Ti nanoparticles with a nonequilibrium face center cubic (FCC) structure from titanium tetrachloride (TiCl_4_) via a capacitively coupled plasma (CCP) route. Furthermore, we demonstrate a secondary H_2_ treatment plasma as an effective strategy to improve the air stability of a thin layer of nanoparticles by further removal of chlorine from the particle surface. Hexagonal and cubic-shaped Ti nanocrystals of high purity were maintained in the air after the secondary H_2_ plasma treatment. The FCC phase potentially originates from small-sized grains in the initial stage of nucleation inside the plasma environment, which is revealed by a size evolution study with variations of plasma power input.

## 1. Introduction

Titanium and its alloys are crucial resources in aerospace and biomedical materials due to their light weight, superior mechanical properties, and biocompatibility [1,2,3]. Besides conventional applications, new functionalities are enabled at the nanoscale, including catalytic activity [4], superplasticity [5], and hydrogen storage capacity [6]. Zhang et al. [4] successfully synthesized ultrasmall metallic Ti nanoparticles (3–5 nm), which demonstrated a high catalytic activity for hydrogen storage reactions at low temperatures. Ti-based alloys with an in situ-formed nanostructured dendritic phase in a nanocrystalline matrix were produced by He et al. [7], exhibiting high plastic strains, high ductility, and a low catastrophic failure rate. Pure Ti under thermodynamic equilibrium exhibits a hexagonal close-packed (HCP, α-Ti) structure at room temperature and transforms to a body-centered cubic (BCC, β-Ti) structure at elevated temperatures around 882 ℃ [8]. Face-centered cubic (FCC, γ-Ti) titanium is a metastable phase in the bulk, but has been reported in nanoscale thin films, interfaces, and particles below critical sizes [9,10,11,12].

While the industrial method named the Kroll process produces Ti sponges of high purity, this process involves energy-intensive extraction and refining procedures requiring long processing times and high temperatures, which introduces environmental and economic concerns [13,14]. To overcome these limitations and enable nanoscale Ti production, various techniques have been developed. Maeng et al. [15] employed a gas-phase co-flow flame method to produce Ti particles of 20–100 nm diameter. A thin layer of NaCl was coated onto the surface via combustion of a TiCl_4_—Na mixture, and then removed in vacuum by a post heat treatment at 800 ℃. Tokoi et al. [16] prepared Ti nanopowder via a pulsed wire direct current discharge with a Ti wire as the precursor. An organic vapor environment was generated by electrically heating oleic acid vapor/mist to encapsulate the particles. Mohammadi et al. [17] extracted nanosized Ti by melting and evaporating bulk Ti samples using an electromagnetic levitation technique with energy supplied by a high-power and high-frequency radio frequency (RF) generator. Park et al. [18] utilized an inductively coupled RF plasma to dissociate and reduce titanium tetrachloride with hydrogen, which enabled the production of Ti powders around 200 nm in diameter, yet with strongly prevalent Ti_2_O_3_ or TiO phases. All the approaches reviewed so far, however, still suffer from low throughput and high energy cost. Moreover, developing a feasible technique to protect metallic titanium nanoparticles from oxidation is challenging. Additional encapsulation is often required due to the strong propensity for titanium to oxidize. This is particularly problematic for particles at the nanoscale [16,19], where the high surface-to-volume ratio implies that a large volume of the metal is consumed, forming a thin native oxide. Few studies to date have been able to demonstrate good control over the chemical purity, particle agglomeration, and oxidation characteristics.

Plasmas, as a highly energetic and rapid gas-phase synthesis technique, are paving the way for sustainable, fully electric, and solvent-free synthesis of a variety of functional nanomaterials. Despite extensive research on their applications in elemental and compound semiconductor nanomaterials production [20,21,22,23], few studies have been reported that explore the potential of plasmas for metal nanoparticle syntheses [24,25,26]. Theoretical calculations showed that an atmospheric-pressure thermal plasma with mixture of Ar, H_2_, and TiCl_4_ is a feasible route for the production of titanium nanoparticles [27,28]. The authors argued that a nonequilibrium process favors a high yield of solid Ti particles due to a faster quenching rate. We propose that nonthermal plasmas, with energetic electrons and radicals, present the potential to efficiently dissociate TiCl_4_, and cause the nucleation of metallic Ti nanoparticles. Negative charging on the nanoparticle surface via collisions with electrons may further prohibit agglomeration and enable precise size control [29,30].

Here, we report on the rapid vapor synthesis of metallic Ti nanoparticles from TiCl_4_ in a one-step capacitively coupled plasma (CCP) reactor on timescales of around 20 ms. Interestingly, a nonequilibrium FCC structure was observed for all material produced in the study. A new approach to address the oxidation of thin films of Ti nanocrystals through secondary H_2_ plasma post-treatment was further demonstrated in this work, which allows the removal of residual chlorine species from the nanoparticle surface and thereby improves their robustness to oxidation in air.

## 2. Materials and Methods

### 2.1. Synthesis of Ti NPs

Metallic titanium nanoparticles were synthesized using a radio frequency (RF) nonthermal plasma reactor, as illustrated in Appendix A. The RF power was supplied by a Tektronix AFG 3251 (Tektronix, Beaverton, OR, USA) arbitrary functional generator generating a sinusoidal waveform of 13.56 MHz amplified by an ENI A150-1210 amplifier (Electronics & Innovation, Ltd., Rochester, NY, USA). A Vectronics HFT1500 (Vectronics, Starkville, MS, USA) matching network connected to the power supply served to transmit the power to a pair of copper ring electrodes to ignite the plasma in a quartz tube with a length of 19.1 cm, 3.7 cm outer diameter, and 3.6 cm inner diameter. The powered electrode was positioned 0.6 cm from the upstream argon injection port, labeled Ar_up_. The spacing between the powered electrode and ground electrode was 1.9 cm.

Titanium tetrachloride (99.995%, Neta Scientific, Inc., Hainesport, NJ, USA) was used as the Ti precursor. TiCl_4_ is a volatile liquid with a vapor pressure of around 10 Torr at room temperature, which exhibits high reactivity with water in the air to form hydrochloric acid, titanium hydroxide, and titanium oxychlorides. H_2_ was used as Cl scavenger gas. Two argon streams were used in this reactor, one to control the upstream pressure (Ar_up_) and one to carry the vaporized titanium tetrachloride (Ar (TiCl_4_)). TiCl_4_ was vaporized through a bubbler system equipped with a needle valve to achieve independent control of the bubbler pressure and the Ar (TiCl_4_) flow rate to tune the vaporized TiCl_4_ concentration in the carrier gas. The bubbler was placed in a water–ice bath to maintain a stable temperature at 0 ℃, at which the vapor pressure of TiCl_4_ is around 2.1 Torr [31]. H_2_ and Ar (TiCl_4_) were injected into the plasma through an inner tube located at 0.6 cm from the Ar_up_ injection port. Particles were synthesized under the conditions listed in Appendix A. Nanoparticles were collected by inertial impact deposition onto glass substrates through an orifice with a length of 5 mm and width of 0.4 mm downstream from the chamber [32].

For the H_2_ plasma post-treatment of the as-synthesized Ti nanoparticles, a secondary capacitively coupled needle-shaped copper electrode was placed 0.5 cm next to the substrate to ignite the hydrogen plasma to remove residual chlorine species. Particles were treated by the secondary plasma with 180 sccm H_2_ flow and 25 sccm Ar_up_ at 150 W for 5 min. The chamber pressure was maintained at around 2.5 Torr. This step aims to reduce the rapid oxidation of the chlorine-terminated nanoparticle surface and maintain the original crystalline structure of the thin layer of Ti nanoparticles deposited on the transmission electron microscope (TEM) grids for the characterization.

### 2.2. Materials Characterization

X-ray photoelectron spectroscopy (XPS) was conducted on a PHI Versa Probe III XPS and UPS (UV Photoelectron spectroscopy) (ULVAC-PHI, Inc., Kanagawa, Japan) system using an Al Kα source. XPS survey scans were taken at a bandpass energy of 280 eV for 10 scans to determine the chemical components. A 55 eV band pass energy with 10 scans was used to collect high-resolution spectra to determine the chemical bonding states of titanium, oxygen, chlorine, and carbon elements. Identification and fitting of the peaks were conducted using the PHI’s “Multipak” software, https://www.multipak.com.tr/. The adventitious carbon peak located at 284.8 eV was used to shift and calibrate the spectra.

X-ray diffraction (XRD) patterns were obtained using a Bruker D8 Discover diffractometer system (Bruker, Billerica, MA, USA) with a cobalt source and a beryllium area detector. The collected patterns were mathematically converted to copper source patterns. Average crystal sizes were estimated using the Scherrer equation [33],
$$L=\frac{K\lambda }{\beta \cos\theta },$$
where K is the Scherrer constant, L is the mean size of the ordered crystalline domain, β is full width at half maximum of the diffracted beam, λ is the wavelength of the incident X-rays, and θ is the Bragg angle.

A Thermo Fisher Talos FX200 (scanning) transmission electron microscope (STEM) (Thermo Fisher Scientific Inc., Waltham, MA, USA) equipped with a Super-X energy-dispersive X-ray (EDX) detector operating at 200 kV accelerating voltage was used for the acquisition of conventional TEM (CTEM) images, selective area electron diffraction (SAED) patterns, high-angle annular dark-field (HAADF) images, and energy dispersive spectroscopy (EDS) elemental mapping. The particle size distribution was determined by manually measuring 300 particles in the TEM micrographs. A log-normal fit was used to quantify the distributions in terms of their respective geometric means and standard deviations. 

Ultraviolet–visible (UV-Vis) measurements were performed using a Cary 7000 UV-Vis spectrophotometer (Agilent, Santa Clara, CA, USA) with a Tungsten halogen visible and deuterium arc UV light source. The nanoparticle films were deposited onto glass substrates. The background of the glass substrate was subtracted for each measurement.

## 3. Results and Discussion

### 3.1. Characterization of Ti Nanoparticles

Powder samples of Ti nanoparticles were characterized by performing X-ray photoelectron spectroscopy measurements to examine the chemical purity of as-synthesized nanoparticles, as depicted in Figure 1a. Deconvolution of the Ti spectra resulted in three doublets (2p_3/2_ and 2p_1/2_). The predominant 2p_3/2_ peak at 454.5 eV arose from metallic state Ti^0+^, along with two peaks at 456.6 eV and 458.8 eV corresponding to different oxidation states Ti^3+^ and Ti^4+^, respectively [34,35,36]. According to the integrated intensity, the ratio of the different valence states Ti^0+^:Ti^3+^:Ti^4+^ is quantified to be 66%:24%:10%. O 1s and Cl 2p spectra were also collected to examine the oxidation level and residual chlorine content, as shown in Appendix A. The O 1s peak at 530.0 eV corresponds to metal oxide ion O^2−^. The other peak at 532.2 eV is attributed to hydroxide (OH^−^) [37,38]. The Cl 2p doublet with 2p_3/2_ at 198.5 eV and 2p_1/2_ at 200.5 eV are assigned to Cl^−^ bonded to Ti [39]. Overall, the as-synthesized nanoparticle powder exhibited a high purity of 66% metallic states with a certain degree of natural oxidation on the surface. 

XRD was utilized to reveal the crystal structure and size information of powder samples. The pattern for the as-synthesized nanoparticles, shown in Figure 1b, was indexed to the nonequilibrium FCC crystal structure with the reflections located at around 38.2°, 44.4°, 64.7°, and 77.7° 2θ ascribed to the (111), (200), (220), and (311) crystal planes of the FCC structure [40,41]. The crystal size was estimated to be around 48 nm using the (111) peak to fit the Scherrer equation [33]. The absence of any crystalline oxide (TiO_2_) peaks suggests that any oxide formed is limited to thin amorphous layers on the surfaces of the metallic particles. The lattice parameter calculated from the peak positions is a=0.407 nm, which agrees well with the prediction from the first-principle calculations [42,43,44].

To understand the particle morphology and size distribution, TEM measurement with size distribution statistics was carried out, as illustrated in Figure 1c,d. TEM samples use sub-monolayer deposits of Ti nanoparticles that are likely to be much more prone to oxidation than the powder samples studied by XPS and XRD because the nanoparticle surfaces of the TEM samples are directly exposed to the atmosphere. In contrast, in powder samples, water vapor and oxygen have to diffuse through nanometer-scale voids to reach nanoparticles below the first few particle layers, which should significantly slow down the oxidation of particles deeper within the sample. Spherical particles are observed with a geometric mean size of 64.6 nm. The geometric standard deviation of 1.15 suggests that the as-deposited nanoparticles are monodispersed with a narrow size distribution. It is noted that the particle size from TEM measurement is larger than the crystal size determined from XRD peaks. We assume that this discrepancy is a result of natural oxidation of the Ti nanoparticles surfaces, which leads to the formation of an amorphous oxide shell at the expense of a reduced Ti crystalline core. In fact, Figure 1c indicates that the oxidized surfaces are amorphous, as only a small core of 10 nm with crystalline lattice fringes (d = 0.24 nm, FCC, (111)) is recognized. 

The surface states of nanoparticles are well-documented to strongly influence their oxidation behavior [45,46]. A comparative study of the synthesis of silicon nanocrystals from silane and silicon tetrachloride precursors showed that nanoparticles with a chlorine terminated surface are more prone to oxidize in air compared to particles with a hydrogen terminated surface [46]. There are two potential oxidation pathways in our as-synthesized nanoparticles after being exposed to the air. Firstly, oxygen molecules in the air can react with metal atoms on the particle surface to form oxide layers. Secondly, the residual chlorine termination on the surface can rapidly react with water vapor in the air to form oxide through the hydrolysis reaction approximated as ${\mathrm{TiCl}}_{4}+2H_{2}O\to {\mathrm{TiO}}_{2}+4\mathrm{HCl}$. The latter is a faster process and we propose this reaction to be dominant for the as-synthesized nanoparticles that were shown to have a chlorinated surface.

To better understand the compositional variation in the core-shell particles, high-resolution HAADF characterization with elemental mapping, and line profile and area profile EDS were performed, as shown in Appendix A. The maps in Appendix A indicate that Ti is more concentrated in a small core region while O is in the shell part. Quantification of the elements is realized by EDS and line profiling. Three areas of interest are chosen from the HAADF image to produce the EDS spectra, as shown in Appendix A, area 1 at the background grid, area 2 at the crystalline core, and area 3 at the amorphous shell. Line profiling is performed across the particle, see Appendix A. It can be learned that the atomic ratio of Ti:O increases from 15%:35% in the amorphous shell region to around 25%:30% in the crystalline core region, indicating lower oxidation and higher purity in the core, which could be consistent with a Ti core surrounded by a thick Ti oxide shell, as will be discussed in more detail below. Cl constitutes around a 5% atomic fraction of the particle and is more concentrated in the core. The actual Cl percentage in the original unoxidized particles before exposure to the air could possibly be higher than this value, as the hydrolysis reaction with water vapor in the air will form HCl vapor to release the residual Cl.

### 3.2. Secondary H_2_ Plasma Treatment

To tackle the oxidation issue of the Ti nanoparticle deposits upon exposure to air, a secondary H_2_ plasma was applied to post-treat the as-deposited nanoparticles in the vacuum chamber, as illustrated in the downstream part in Appendix A. It is hypothesized that the hydrogen radials generated during this process will react with surface chlorine through the pathway H+Cl→HCl and be pumped away, thus reducing the contribution of the rapid chlorine-mediated oxidation of the Ti nanoparticle surface. Figure 2a represents the CTEM image of a collection of Ti nanoparticles after the treatment. The average size is around 65 nm with a narrow distribution, which is consistent with the particles produced without the secondary treatment. It can be thus inferred that the hydrogen treatment is more of a surface chemical process rather than a thermal process since no phase or size change is observed. However, the structure of the particles is notably different, with large metallic cores and only thin oxide shells. The SAED pattern from the collection of particles (see Figure 2b), shows diffraction rings yielding interplanar distances of 0.24 nm, 0.21 nm, 0.15 nm, and 0.13 nm, which match the FCC (111), (200), (220), (311) planes, respectively. It should be noted that the as-treated particles consist of cubic and hexagonal shapes, which are different from the spherical particles in Figure 1c. Higher magnification views of the representative cubic and hexagonal shaped nanoparticles are displayed in Figure 2c,d. Lattice spacings of 0.21 nm (FCC, (200) crystal plane) and 0.24 nm (FCC, (111) crystal plane) are identified, see inset of Figure 2c,d, through the fast Fourier transform (FFT) and inverse FFT image processing. 

Both cubic and hexagonal particles display core-shell structures evident in the distinct Z contrast in Figure 2c,d. Elemental mapping of these particles further confirms that O and Cl are mainly located at the particle surface while Ti is at the core, as depicted in Figure 3a and Figure 4a. In the cubic particle, O is confined to a thin shell while Cl is negligibly in the core and at noise level across the whole frame. In the hexagonal particle, O and Cl mostly reside in a thicker shell, which suggests a possibly stronger diffusion into the core than the cubic particle. 

Radial elemental distributions are further quantified by line profiling across the particles in Figure 3c and Figure 4c. Given that the intensity counts are in effect an integration from both the core and shell regions projected along the electron beam axis, deconvolution of the signals into a core and a shell chemical composition is required. A cubic core-shell model is structured to predict the core and shell radii and chemical composition of the cubic nanoparticle, as demonstrated in Figure 3b. For simplification, a spherical shape is assumed to approximate the hexagonal nanoparticle, see Figure 4b. Details about the modeling can be found in the Appendix A. The line counts are assumed to be proportional to the electron beam scanning lengths in the core, denoted as $L_{c},$ and the shell, denoted as $L_{s}$. $L_{c}$ and $L_{s}$ are functions of the core radius R_c_ and shell radius R_s_ depending on the distance from the center of the particle p. The fitted parameters are summarized in Appendix A. 

For the cubic particle, a core of pure Ti and a shell of Ti_0.52_O_0.40_Cl_0.08_ yields the optimized fit with the experimental profile. The core radius is determined to be $R_{c}=27\;\mathrm{nm}$ from the O profile. The Cl profile is best fit with a shell with an inner radius of 31 nm. By subtraction from the particle radius $R_{s}=37\;\mathrm{nm}$, it was found that O is distributed in a shell thickness of 10 nm while Cl is distributed in a shell thickness of 6 nm, which is visualized in Figure 3d. To assess the goodness of fit, normalized root mean square errors are calculated to be 0.68, 0.59, and 1.09 for the fitting of Ti, O, and Cl, respectively, included in the inset of Figure 3c. 

By approximating the hexagonal particle with a spherical shape, a pure Ti core and a shell of Ti_0.56_O_0.31_Cl_0.13_ are found to best fit the profiles. The particle radius is 46 nm, while the core radius deduced from the O profile is 34 nm and from the Cl profile is 29 nm. Based on the fitted parameters, Figure 4d represents the modeled core-shell structure with an essentially pure Ti core, an O-containing shell with a thickness of 12 nm, and a Cl-containing shell with a thickness of 17 nm. Compared to the hexagonal-shaped particle, the cubic nanoparticle shows a larger portion of a pure Ti core, a narrower O-containing shell, and lower Cl content, which indicates more thorough Cl removal by the H_2_ plasma treatment. Additionally, EDS area profiles from the particle center (area 1) and particle periphery (area 2) provided in Appendix A further confirm the purity of the particle core with the absence of Cl peaks. 

We cannot rule out the possibility that additional energy input from the secondary H_2_ plasma further enhances particle crystallinity with the existence of energetic electrons and radicals. To test whether the hydrogen plasma could reverse the oxidation of Ti nanoparticles, such as those shown in Figure 1, Ti nanoparticles were collected onto a TEM grid, exposed to the air for one minute, and then transferred back to the vacuum chamber for H_2_ treatment under the same conditions. Appendix A shows the TEM images of the oxidized particles before and after the H_2_ treatment. Little change in the particle morphology obtained before and after the H_2_ plasma treatment was observed. This demonstrates that H_2_ plasma is not reducing or recrystallizing the oxidized particles but that its effect is mainly in the removal of Cl post synthesis. It can be hypothesized that the original cubic and hexagonal nanoparticle surfaces seen in Figure 2 are roughened and evolve to spherical ones with the subsequent oxidation of the chlorine-terminated surface in the air, as seen in Figure 1.

The optical absorption properties of the post-treated nanoparticle films were also examined in this work. Metallic nanoparticles usually possess localized surface plasmon resonance (LSPR) due to the coherent collective oscillation of delocalized electrons in response to the electromagnetic field of a certain wavelength. Metallic Ti will yield a plasmonic resonance in the visible wavelength at around 400–500 nm, while oxidized TiO_2_ will absorb in the UV range below 400 nm due to the wide bandgap (3.0–3.2 eV) [47,48,49]. UV-vis absorption spectra were measured to examine the optical absorption of the nanoparticle films, as depicted in Appendix A. An LSPR band is visible at around 450 nm for the H_2_ plasma-treated nanoparticle film, which corroborates the stability of the treated nanoparticles in the air. In the untreated control sample, this feature is not present, presumably due to the much stronger oxidation of the Ti nanoparticles.

Compared with untreated amorphous nanoparticles in Figure 1, the H_2_ plasma-treated sample shows a high degree of crystallinity and purity with shapes of higher surface energy as well as plasmonic features. It can be thus concluded that H_2_ plasma is a viable strategy to remove surface chlorides from the as-synthesized Ti nanoparticles to reduce their subsequent oxidation and maintain the crystallinity and purity of particles. A detailed study of the secondary H_2_ plasma parameters is of great interest but beyond the scope of this study.

### 3.3. Size Evolution for FCC Phase Formation

Next, we discuss the possible mechanism for the formation of the FCC phase. We first excluded the potential formation of a cubic titanium hydride (TiH_x_) based on the lattice parameter determined by XRD since TiH_x_ would have a lattice parameter ~0.45 nm, while the results showed a lattice parameter of ~0.41 nm consistent with metallic Ti [50,51]. Xiong et al. [10] derived the size-phase diagrams for various Ti nanostructures and explored the stability ranges of the HCP, FCC, and BCC phases. The authors found that, for a temperature of 600 K, the FCC structure is stable for Ti nanoparticles smaller than ~6.5 nm diameter and transitions to the HCP structure for larger particles. The FCC structure remained stable for even larger Ti particles at higher temperatures, up to a maximum of ~27 nm diameter at 1156 K. While these results may be modified for our conditions due to the Cl surface species that likely affect the surface energies in our synthesis, they may still offer a clue as to the presence of the FCC phase observed here.

Particle growth in plasmas is usually associated with three different growth phases [52,53]: (1) dissociation of the precursor and nucleation of initial clusters, (2) agglomeration of these clusters into larger particles that at some point become large enough to become uniformly negatively charged so that Coulomb repulsion prevents further agglomeration, and (3) continued growth of these nanoparticles through surface deposition of precursor fragments. To explore different growth stages, we here use plasma power as an admittedly imperfect parameter to control particle growth, with larger powers presumably leading to more TiCl_4_ dissociation and, thus, faster growth of Ti nanoparticles. Figure 5a–d show Ti particles grown at different powers. Figure 5e shows the particle size distributions determined from the primary particles seen in Figure 5a–d. It is noted that the agglomerates observed in Figure 5a–c likely form after particles leave the plasma and that Figure 5e displays the size distribution of the constituent particles [53]. The geometric mean of the particle size increases from 14.5 nm to 64.6 nm with power increasing. However, the geometric standard deviation decreases from $\sigma_{g}=1.31$ at 80 W to $\sigma_{g}=1.15$ at 150 W. In aerosol dynamics, this reduction in the geometric standard deviation is associated with the end of the cluster agglomeration phase and the transition to the surface growth phase [54]. 

Based on this analysis, it appears that larger Ti nanoparticles are likely the result of surface growth on seed crystals that may have formed by initial cluster nucleation and agglomeration. Based on the study by Xiong et al. [10], these seed crystals may have formed in the FCC phase as the thermodynamically preferred phase for small particles. It is conceivable that the FCC structure is preserved as small-sized grains continue to grow through epitaxial deposition of TiCl_x_ fragments on the particle surfaces. Further study of this aspect is required in the future.

## 4. Conclusions

To summarize, the formation of metallic FCC Ti nanoparticles from TiCl_4_ was experimentally investigated in a flow-through, single-step, capacitively coupled plasma reactor with Ar as the carrier gas and H_2_ as the chlorine scavenger. As-produced Ti nanoparticles are highly oxidized. We developed a secondary H_2_ plasma treatment technique that reduced the oxidation and improved the air stability of the Ti nanoparticles, likely by scavenging residual Cl from the Ti nanoparticle surfaces. This approach may hold promise for other energetic metal nanomaterials with high oxygen affinity. Through analysis of the particle size distribution, we determined that larger particles are grown through surface deposition on initial smaller seed crystals. For such small seed crystals, the FCC phase may be the thermodynamically preferred phase, and epitaxial deposition on the seed crystals may preserve this phase also in larger particles. Further work needs to be performed to better understand the formation of the FCC structure in the plasma environment. Overall, the nonthermal plasma method presented here may be a fully electric, all-gas-phase approach for more sustainable production of Ti from TiCl_4_.

## Figures and Tables

**Figure 1 nanomaterials-14-00264-f001:**
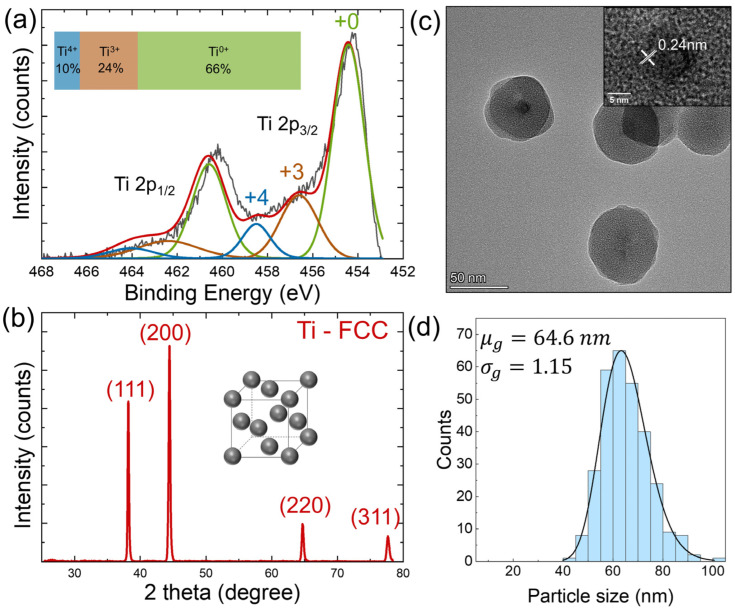
Characterization of as-synthesized metallic Ti nanoparticles prior to hydrogen plasma post-treatment. (**a**) XPS high-resolution spectrum of Ti 2p, inset shows the ratio of different valence states according to integrated intensities. (**b**) XRD pattern of the as-synthesized nanoparticles. (**c**) CTEM images of as-synthesized nanoparticles; inset shows zoomed-in view of lattice fringes of 0.24 nm corresponding to FCC (111) crystal plane. (**d**) Size distribution statistics determined by TEM of 300 particles with geometric mean and standard deviation of the lognormal fit.

**Figure 2 nanomaterials-14-00264-f002:**
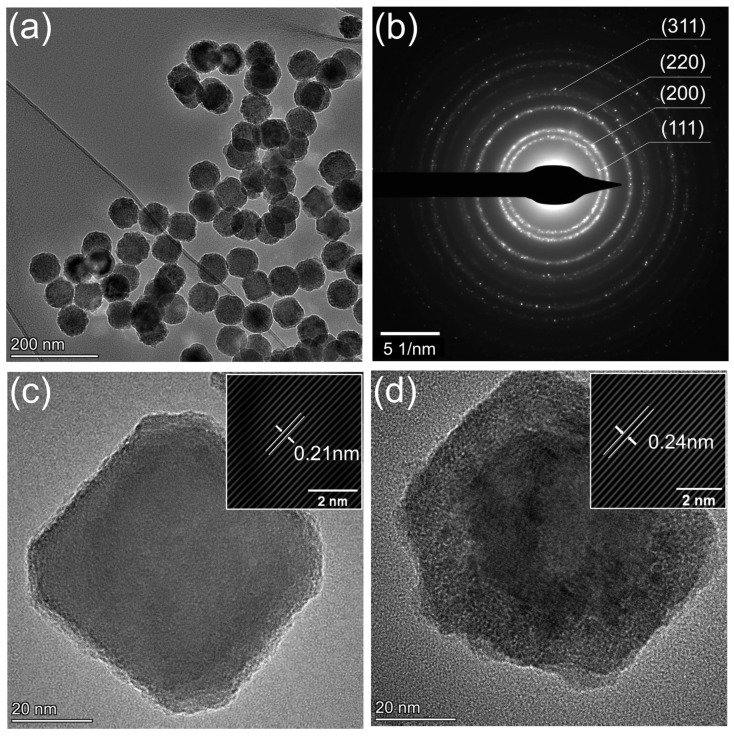
(**a**) CTEM images of Ti nanoparticles after the secondary H_2_ plasma treatment (**b**) SAED pattern from the collection of Ti nanoparticles (**c**) HRTEM of cubic shaped nanoparticle (**d**) HRTEM of hexagonal shaped nanoparticle.

**Figure 3 nanomaterials-14-00264-f003:**
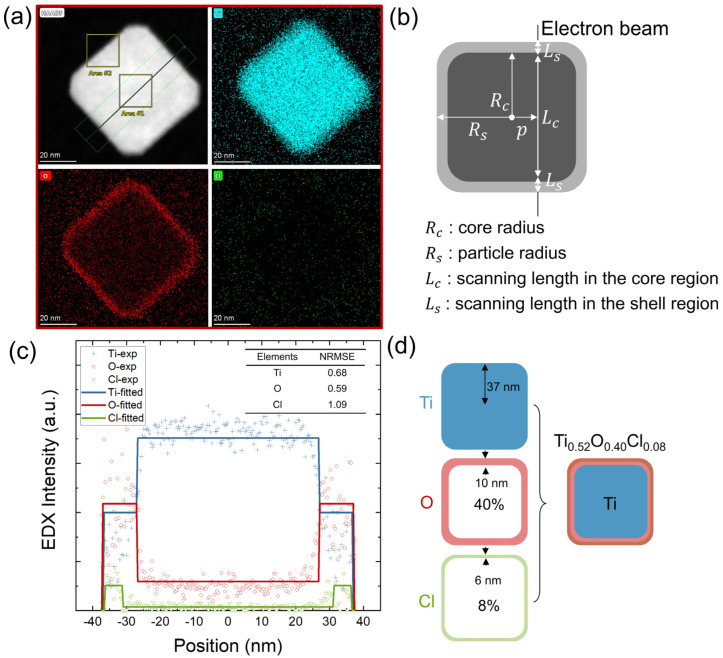
HAADF-STEM-EDS mapping images of cubic-shaped nanoparticles treated with a secondary H_2_ plasma (**a**) HAADF and EDS mapping images (**b**) schematic of the model of core-shell spherical nanoparticle (**c**) measured and fitted line scan profiles across the particle, with calculated NRMSE in inset table (**d**) schematic of the fitted core-shell nanoparticle.

**Figure 4 nanomaterials-14-00264-f004:**
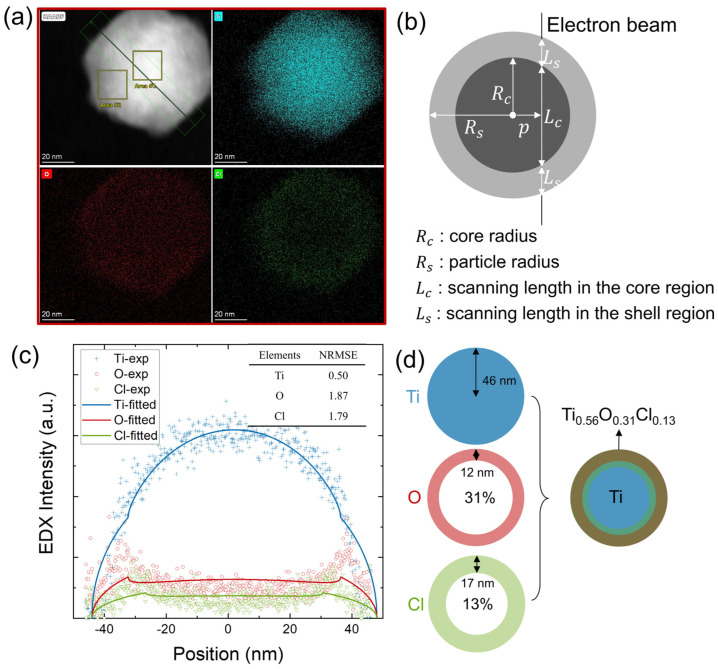
HAADF-STEM-EDS mapping images of hexagonal-shaped nanoparticles treated with a secondary H_2_ plasma (**a**) HAADF and EDS mapping images (**b**) schematic of the model of core-shell spherical nanoparticle (**c**) measured and fitted line scanning profile across the particle, with calculated NRMSE in inset table (**d**) schematic of the fitted core-shell nanoparticle.

**Figure 5 nanomaterials-14-00264-f005:**
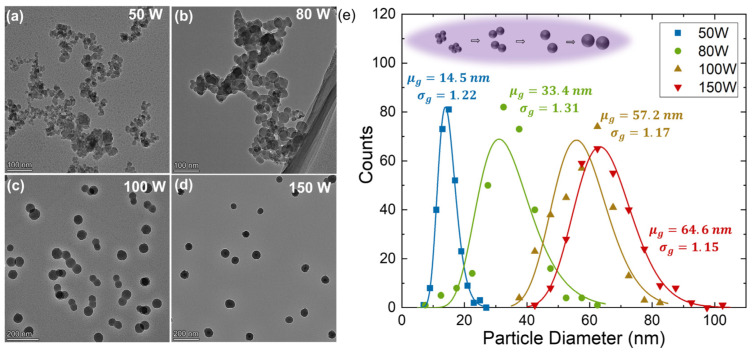
CTEM images of as-synthesized nanoparticles with plasma power of (**a**) 50 W (**b**) 80 W (**c**) 100 W (**d**) 150 W and (**e**) size distribution statistics from 300 particles with geometric mean and standard deviation of the lognormal fit.

## Data Availability

Data are contained within the article and Appendix A.

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
