# Peer review of "Nonthermal Plasma Synthesis of Metallic Ti Nanocrystals"

_nanomaterials, 2024, doi:10.3390/nano14030264_

Round 1
Reviewer 1 Report
Comments and Suggestions for Authors
This manuscript is to prepare fcc-structured Ti nanoparticles using a facile nonthermal plasma technique along with advanced characterizations, including XPS, XRD, and TEM analyses. Some minor corrections should be considered as follows:
1. Discussing the potential nanoscale Ti applications would help understand the significance of the synthesis method. Also, future studies based on this synthesis and analysis should be added to the conclusion.
2. Rationalization of this discussion (“TEM samples use sub-monolayer deposits of Ti nanoparticles that are likely much more prone to oxidation than the powder samples studied by XPS and XRD.” (Lines 157 – 159)) should be added.
3. Is there any fundamental reason for forming the core-shell structure? What is the reason for selecting the (111) peak to fit the Scherrer equation to estimate the particle sizes? Are there other Ti precursors to improve the oxidation problem rather than TiCl4?
4. The inserts in Figure 2c,d should be revised as they are unclear. The scale bar in Figure 2b should be corrected from 5 nm-1 to 5 nm. The XRD hkl (311) index (Line 150) is not the same as that in Figure 1b ((311) versus (310)). Also, the (311) plane (Line 217) should be confirmed. The Cl-related value in Figure 3d should be consistent with that in text (Line 217, 6 nm versus 5 nm).
Comments on the Quality of English LanguageNone
Reviewer 2 Report
Comments and Suggestions for Authors
In this manuscript, high-purity titanium nanoparticles with non-equilibrium face center cubic (FCC) structure were prepared by using TiCl4 precursor. The stability of Ti NPs in air was improved by removing of chlorine (CI) on the surface with secondary hydrogen. It has great application potential in the fields of aerospace and biomedicine. However, there are still the following shortcomings:
(1) It is an important mean to treat plasma with secondary hydrogen. What are the mixing ratio and pressure values of H2 and Ar gases? What is the basis for setting the flow rates of 25 sccm and 180 sccm in Table S1? Furthermore, what is the basis for setting the power parameter to 150 W and the time to 5 min during the secondary H2 plasma treatment?
(2) Whether the working condition is in vacuum, when the hydrogen plasma is ignited to remove chlorides? If not, the reaction equation should be TiCl4 + 2H2 + O2 = TiO2 + 4HCl. Next, the research only considers the influence of chloride on Ti NPs, but how to consider for TiO2?
(3) When analyzing the spin-orbit peaks in Section 3.1, what are the ratios of the areas of Ti0+, Ti3+, and Ti4+ at the 2p1/2 and 2p3/2? Is it approximately 1:2? Similarly, what is the ratio of CI in the 2P electrons peaks? These data can be characterized more accurately by XPS. Meanwhile, this study mainly uses the combination of Scherrer equation and TEM to characterize the sizes of Ti NPs. Please give the specific definition of Scheller equation with reference literatures.
(4) What is the reason for Ti NPs changing from spherical to cubic and hexagonal shapes? What is the morphological influence on its properties?
(5) What is the light source when performing UV-Vis analysis in Section 3.2?
(6) The whole paper is aimed at obtaining FCC phase Ti NPs. What are the merits compared with HCP and FCC structures? At the same time, what are the best conditions and mechanisms for formation of FCC structure?
(7) First person pronoun, such as "we" and "our", appear several times in manuscript, should be eliminated.
